# Crowdsourcing airway annotations in chest computed tomography images

**Veronika Cheplygina**[1]*, **Adria Perez-Rovira**[2,3], **Wieying Kuo**[3,4], **Harm A. W. M. Tiddens**[3,4], **Marleen de Bruijne**[2,5]

**1** IMAG/e, Department of Biomedical Engineering, Eindhoven University of Technology, Eindhoven, The Netherlands, **2** Biomedical Imaging Group Rotterdam, Department of Radiology and Nuclear Medicine, Erasmus MC - University Medical Center Rotterdam, Rotterdam, The Netherlands, **3** Department of Pediatric Pulmonology and Allergology, Erasmus MC - Sophia Children's Hospital, Rotterdam, The Netherlands, **4** Department of Radiology and Nuclear Medicine, Erasmus MC - University Medical Center Rotterdam, Rotterdam, The Netherlands, **5** Machine Learning Section, Department of Computer Science, University of Copenhagen, Copenhagen, Denmark

* v.cheplygina@tue.nl

**Data Availability Statement:** All data and code are available via Github, http://github.com/adriapr/crowdairway.git.

## Abstract

Measuring airways in chest computed tomography (CT) scans is important for characterizing diseases such as cystic fibrosis, yet very time-consuming to perform manually. Machine learning algorithms offer an alternative, but need large sets of annotated scans for good performance. We investigate whether crowdsourcing can be used to gather airway annotations. We generate image slices at known locations of airways in 24 subjects and request the crowd workers to outline the airway lumen and airway wall. After combining multiple crowd workers, we compare the measurements to those made by the experts in the original scans. Similar to our preliminary study, a large portion of the annotations were excluded, possibly due to workers misunderstanding the instructions. After excluding such annotations, moderate to strong correlations with the expert can be observed, although these correlations are slightly lower than inter-expert correlations. Furthermore, the results across subjects in this study are quite variable. Although the crowd has potential in annotating airways, further development is needed for it to be robust enough for gathering annotations in practice. For reproducibility, data and code are available online: http://github.com/adriapr/crowdairway.git.

## Introduction

Chest computed tomography (CT) can be used to quantify structural abnormalities in the lungs, such as bronchiectasis, air trapping and emphysema, which in turn can be used for diagnostic or prognostic purposes. For example, the airway-to-artery ratio (AAR) is an objective measurement of bronchiectasis which is sensitive to detect early lung disease [1, 2]. Other promising measurements are the wall-area percentage (WAP) and the wall thickness ratio (WTR) which characterize the ratio of the airway wall to the airway lumen [3]. Unfortunately, manual measurements of the airways and vessels suffer from intra- and inter-observer

**Funding:** The author(s) received no specific funding for this work.

**Competing interests:** The authors have declared that no competing interests exist.

variation and are time-consuming (8-16 hours per chest CT) [4]. Machine learning techniques such as [5–7] can be an alternative, but may require a large amount of annotated data to be able to generalize to all situations.

In various applications, crowdsourcing has been proposed as an alternative for tasks where annotated data is scarce. Crowdsourcing refers to outsourcing tasks (often referred to as human intelligence tasks or HITs) to a group of online users (often referred to as knowledge workers or KWs). This strategy has also been quite effective in medical image analysis— [8] surveys over 50 papers where results have been mostly positive. One of these papers is our earlier study [9] where we described our experiences with crowdsourcing airway measurements. We found that 67.8% of the collected results were not valid, i.e. the airway measurements could not be extracted. However, after filtering out such results, strong correlations between the crowd and expert were observed. Although these experiences were encouraging, they only concerned a single chest CT image, and it was unclear whether they could be generalized to other scans.

In this paper we describe crowdsourced airway measurements collected shortly thereafter for a larger set of 24 chest CT images, and with a slightly updated crowdsourcing procedure. With this follow-up study we aim to answer the following questions:

- Does the crowd create valid results?

- What is the quality of the crowd compared to a trained expert, after combining different results per task?

- Can we predict the quality of the crowd results, given a particular scan?

## Materials and methods

### Chest CT scans

We used inspiratory pediatric CT scans from a cohort of 24 subjects [10, 11], collected at the Erasmus MC—Sophia Children's Hospital. These scans have been collected and anonymized for previous studies, and approved to be used for further research. The anonymized scans were shared with us, including the age and sex of the subject, whether the subject had cystic fibrosis, and several measures related to lung capacity and airway size.

The voxel size was $0.5508 \times 0.5508 \times 0.6000$ mm. Each scan contained a number of airways of different generations from the 2nd to the 14th. The most common generations were the 6th (23.7%), 7th (20.0%), 5th (17.7%) and 8th (13.9%).

In each scan, a number of airways were annotated by an expert. The expert localized an airway, outlined the airway lumen (inner airway boundary) and airway wall (outer airway boundary) in a plane approximately perpendicular to the airway center line, and recorded the measurements of the areas.

### Generating airway images

Fig 1 shows a global overview of our method. The first step is to create a crowdsourcing task for each airway, which requires extracting 2D image slices from a 3D volume. This requires having a 3D location and orientation of the airway. Normally this localization would be done by the expert, however in this study we assume that localization was already done, and focus only on outlining the airway in the image.

More specifically, we used 3D voxel coordinates, at which experts have previously outlined airways using the Myrian™ software. We generated 2D slices of $50 \times 50$ voxels.

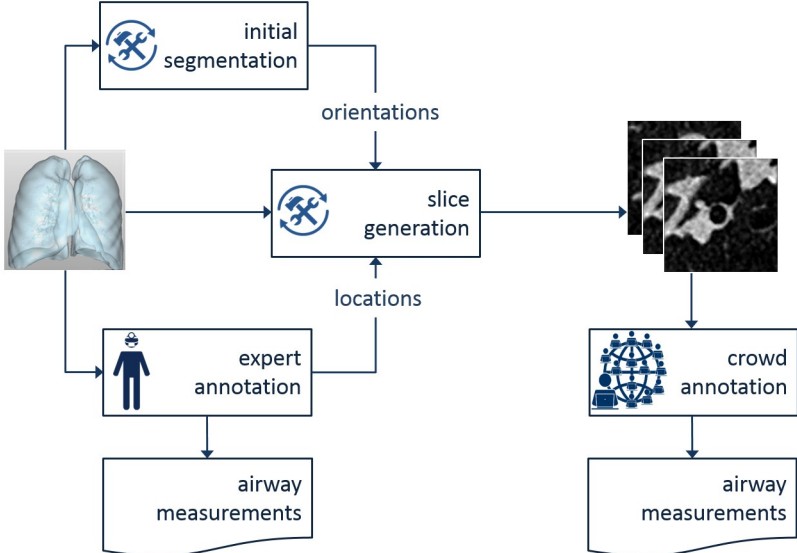

**Fig 1. Overview of the method.** A 3D image is annotated by experts. The locations and orientations of the airways are then used to generate 2D slices of the airways, which are then annotated by the workers.

The slides were reviewed by one observer (APR) to retain only the images with a visible airway that was cut approximately perpendicularly. There were 1026 such images, which are further analysed here. We used cubic interpolation and an intensity range between -950 and 550 Hounsfield units for better contrast, as recommended by the experts. Each image slice was rescaled to 500 × 500 pixels for annotation purposes.

## Annotating airway images

Each of the generated airway images is a crowdsourcing task. A worker assigned to a task creates a result, consisting of one or more annotations (outlines) placed in the image.

To gather these results, we used Amazon Mechanical Turk [12]. All decisions regarding Amazon Mechanical Turk (MTurk) were based on consultation with colleagues who had used MTurk in the past. Apart from updating the instructions to workers, we used the same settings as in our preliminary study, which we repeat here for completeness. All results were collected in 2016.

The annotation interface was integrated into the platform by supplying a dynamic webpage, built with HTML5 and Javascript. This custom-made interface had an ellipse tool, which resembled the tool used by the experts more closely than the default annotation tools available on MTurk. The details of our HIT, which the workers could see when searching for HITs, are shown in Table 1. The workers were instructed to draw two ellipses outlining the airway lumen and the airway wall, or to place a small circle in the top right corner of the image, if no airway is visible. Following our experiences in the preliminary experiments, we revised our

**Table 1. Details of HIT on Amazon Mechanical Turk.**

| Parameter | Value |
| --- | --- |
| Title | Save lives by annotating airways! |
| Description | Draw two contours to annotate an airway (dark circle or ellipse) in image from a lung scan |
| Keywords | image, annotation, contour, draw, drawing, segmentation, medical |

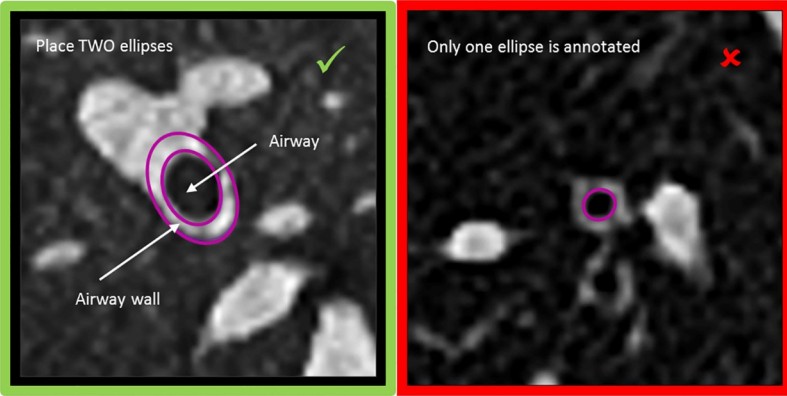

**Fig 2. First set of instructions provided to the workers.** One more set of instructions was available, but this was the set that all workers would see.

instructions, placing more emphasis on the need to draw two ellipses. A screenshot is shown in Fig 2.

We randomly created HITs with 10 images per HIT. A worker could request a HIT, annotate 10 images, and then submit the HIT. The workers were paid $0.10 per completed HIT. Only workers who had previously done at least 100 HITs with an acceptance rate of 90% could request the HITs. We collected 20 results per image, because with 10 results per image as in the preliminary experiment, some images did not have valid annotations.

The data collection was done in 2016, shortly after our preliminary study [9]. Workers consented to doing the task when accepting it on Amazon MT. Due to the non-graphic nature and anonymous character of the airway slices, no other specific approval was obtained. For each result, we recorded an anonymized ID of the worker and the coordinates of the annotations. No other information about the workers was recorded.

## Measuring crowd annotations

We applied a simple filtering step to filter out invalid results. The following results were excluded:

- number of ellipses not equal to 2

- not resized ellipses (default size of the tool is a circle)

- not overlapping ellipses

After filtering, we measured the areas of the inner ($a_i$) and outer ($a_o$) ellipse, and calculated the wall thickness ratio (WTR) and wall area percentage (WAP). The WTR is the wall thickness divided by the outer diameter:

$$WTR = \frac{WT}{d_o} = \frac{(d_o - d_l)/2}{d_o} \tag{1}$$

where $d_o$ is the diameter of the outer ellipse and $d_i$ is the diameter of the inner ellipse, and the wall thickness $WT$ is defined as:

$$WT = \frac{d_o - d_i}{2}. \tag{2}$$

The WAP is the percentage of the total airway area that is airway wall:

$$WAP = \frac{a_o - a_i}{a_o} \times 100. \tag{3}$$

Note that both the workers and the experts place ellipses in slices with roughly perpendicular airways, resulting in almost-circular ellipses. For the workers we have both the major and minor diameters available, but the experts only record the area. For comparisons between the two, we therefore had to assume circular airways.

## Quality of crowd measurements

Before measuring how good the crowd is on each task, we need to combine the results per task. We used three different strategies for this:

Median.  Taking the inner/outer areas of all valid results, and combining them with the median function. WAP and WTR are then calculated based on these median values. This is the strategy used in our preliminary study.

Random.  Selecting a random valid result per task. This gives an indication of how good the crowd could be, if each task was assigned to only one worker, and gives a pessimistically biased indication of how good the crowd could be.

Best.  Taking the valid result that is closest to the expert measurement, based on the inner and outer measurements. This is an optimistically biased indication of how good the crowd could be, if we only selected the best workers.

Additionally, we can choose to exclude tasks that have less than $v$ valid results. This will reduce the number of tasks for which a combined result is available, but will presumably increase the quality of the result.

After combining the results per task, we use the Pearson's correlation coefficient, $\rho$, between the crowd measurement and the expert measurement. Correlation coefficients are interpreted as follows: weak correlation for $0 \leq \rho < 0.3$, moderate correlation for $0.3 \leq \rho < 0.5$, strong correlation for $0.5 \leq \rho < 1$. Note that, if a task has had no valid results, it will be excluded from the analysis.

## Predicting crowd quality

Lastly, we investigate whether any factors contribute to the crowd's performance across different scans in our data. We use the inner airway after median combining as a proxy for the quality.

We then look at the relationship between the quality and the following characteristics:

- Whether or not the subject has cystic fibrosis (CF)

- Forced expiration volume in 1 second (FEV1), which measures how much air a participant can exhale in 1 second.

- Forced vital capacity (FVC), which measures the total volume of air a participant can exhale.

- Number of airways as indicated by the expert.

- Average airway generation, which indicates the number of bifurcations between the current branch and the trachea. Higher generations correspond to smaller airways and vice versa.

We use the Spearman correlation to investigate the relationship of these characteristics, because we cannot assume a linear relationship between them (in particular, the CF status variable is binary). We report the correlation coefficient and the p-value from a two-sided hypothesis test, where the null hypothesis is that the characteristics are not correlated. We use a significance threshold of 0.05. Since we perform five comparisons in total, after adjusting for multiple comparisons the threshold becomes 0.01.

## Results

### Validity of crowdsourced annotations

In total we collected 20520 results for 1026 tasks. A few typical examples are shown in Fig 3. Of these 11742 results (57.2%) were classified as invalid, and 624 (3.0%) contained multiple pairs of ellipses per image, which we excluded to simplify the analysis.

Of the 11742 invalid results, 8809 tasks only had one annotation. This could indicate not seeing an airway, which was the case for 2641 of the results. A further 2933 results had signs of the worker trying to annotate the image (placing ellipses on top of airways), but not following the instructions of outlining two ellipses.

We visually examined several cases where all or most workers indicated not seeing an airway. These appeared to be difficult cases due to low contrast and/or only part of the airway wall being visible (but not so much the size of the airway). While a trained observer would identify these as airways, it is not unexpected that the workers were not able to do so.

The results were created by 577 workers in total, who made as little as 1 or as many as 2313 results. Similar to the observations in [13], most workers only created a few results, and a few workers were responsible for a lot of the results, as shown in Fig 4(a). Fig 4(b) shows the number of valid and invalid results made by each worker. Overall there is a tendency for workers to create more valid than invalid results. However, there are a few workers who have created a lot of results overall, and who tend to create more invalid results. They contribute to 57.2% of the invalid results. Finally, there are no workers that created only invalid results.

### Quality of airway measurements

When considering each result independently (without combining the results per task), there is a correlation of 0.803 for the inner airway and 0.697 for the outer airway.

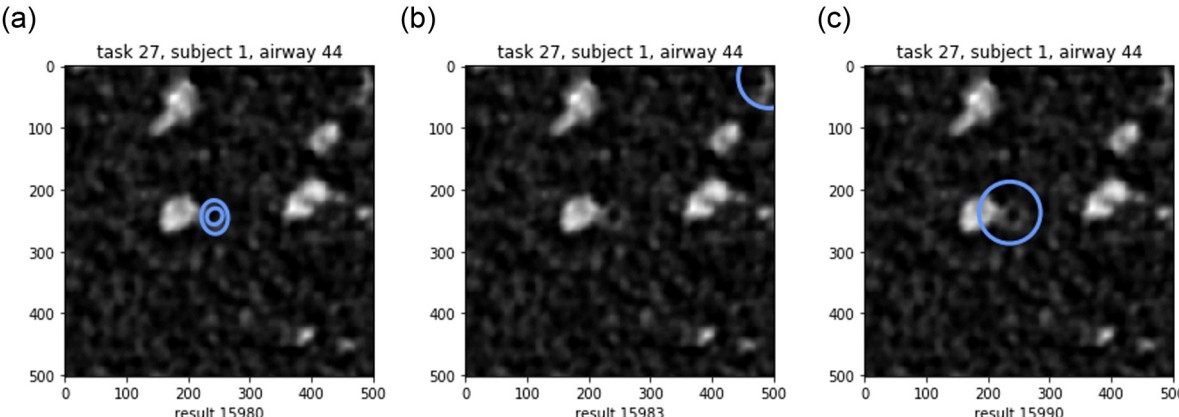

**Fig 3. Example results acquired for the same task: Valid result with two annotations, and two invalid results: A worker who indicates not seeing an airway, and a worker who detects the airway but does not outline it.**

(a)

(b)

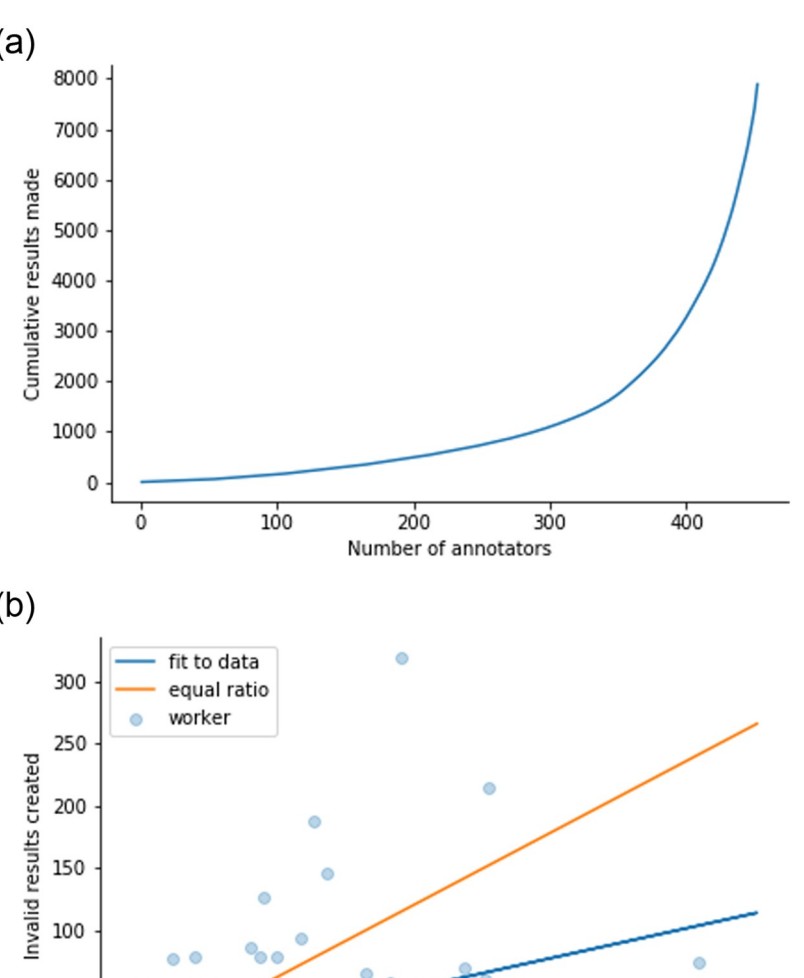

**Fig 4.** (a) Cumulative results made by the workers. Many workers contribute a few results, and a few workers contribute the most results. (b) Valid results vs invalid results by each worker. The blue line shows the fit to the data, and that workers create tend to create more valid results in general.

Additionally, we found moderate correlations for the ratio based measures, 0.426 for the WAP and 0.424 for the WTR.

The airway measurements and correlations after combining the results across workers are shown in Table 2, as well as Figs 5–8. Combining improves all correlations, and for the ratios the correlations can be categorized as strong for median and "best" combining. "Best" combining gives the highest correlations, although the difference with median combining is rather small for the inner airway, WAP and WTR. For the outer airway, the difference is more pronounced (0.769 vs 0.896), suggesting that the task is more difficult, leading to more variation in the crowd.

Overall, since the "best" combining method is optimistically biased due to access to ground truth, our results suggest median combining is a good choice for this data.

**Table 2. Pearson correlations between the expert and the crowd with different combining methods, and between two experts.**

| Method | vs Expert 1 | | | | vs Expert 2 | | | |
|---|---|---|---|---|---|---|---|---|
| | inner | outer | wap | wtr | inner | outer | wap | wtr |
| None | 0.803 | 0.697 | 0.426 | 0.424 | 0.808 | 0.686 | 0.469 | 0.466 |
| Random | 0.803 | 0.701 | 0.421 | 0.418 | 0.813 | 0.679 | 0.489 | 0.481 |
| Median | 0.844 | 0.769 | 0.572 | 0.565 | 0.850 | 0.746 | 0.661 | 0.649 |
| Best | 0.858 | 0.896 | 0.585 | 0.590 | 0.858 | 0.842 | 0.583 | 0.570 |
| Expert 1 | | | | | 0.964 | 0.925 | 0.701 | 0.687 |

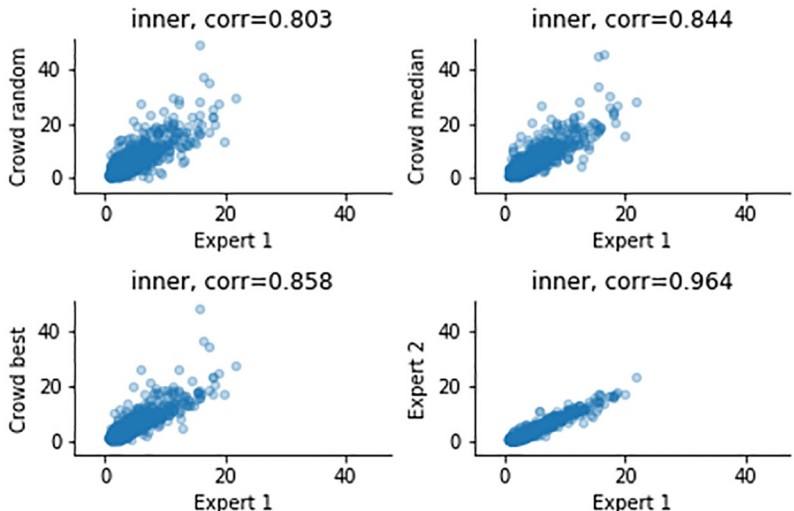

**Fig 5. Measurements of the inner airway, comparing expert 1 (x-axis) and to three combining methods and expert 2 (y-axis).**

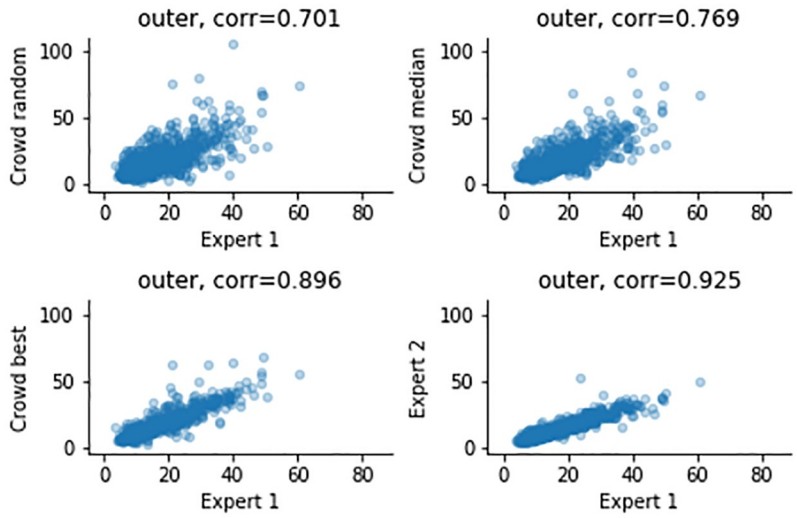

**Fig 6. Measurements of the outer airway, comparing expert 1 (x-axis) and to three combining methods and expert 2 (y-axis).**

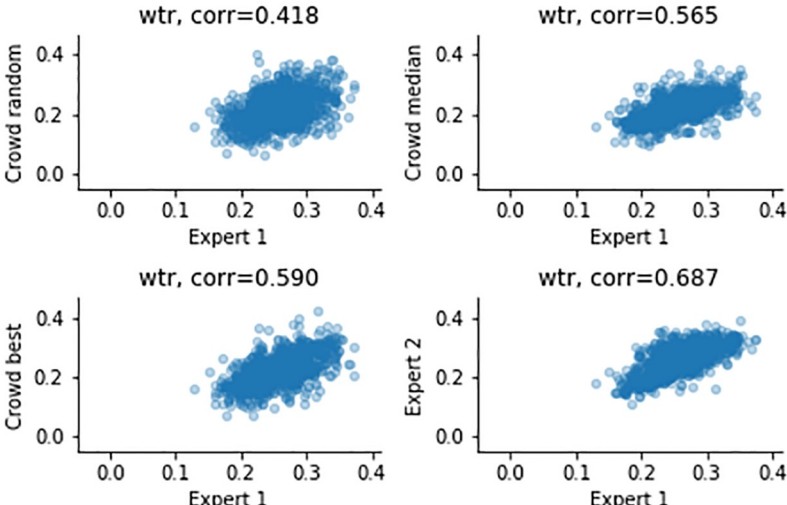

**Fig 7. Measurements of the WTR, comparing expert 1 (x-axis) and to three combining methods and expert 2 (y-axis).**

Median combining simply combines all (between 1 and 20) the valid results available for a particular task. To understand how the number of valid results affects the correlations, we investigated combining only for tasks where at least a certain number of valid results must be available.

The correlations are shown in Fig 9. There is almost no effect on the correlations for the inner and outer airway, and the correlations for WAP and WTR steadily improve as more valid results are combined. This could also indicate that the tasks with more valid results, are in general easier images to annotate.

To summarize, the crowd can create good annotations, and combining annotations using the median helps to improve the quality, although not to the quality of the expert. For median

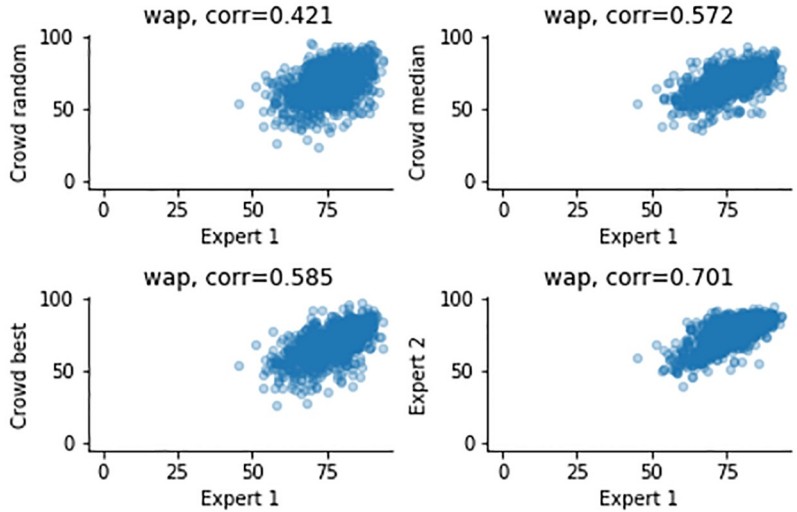

**Fig 8. Measurements of the WAP, comparing expert 1 (x-axis) and to three combining methods and expert 2 (y-axis).**

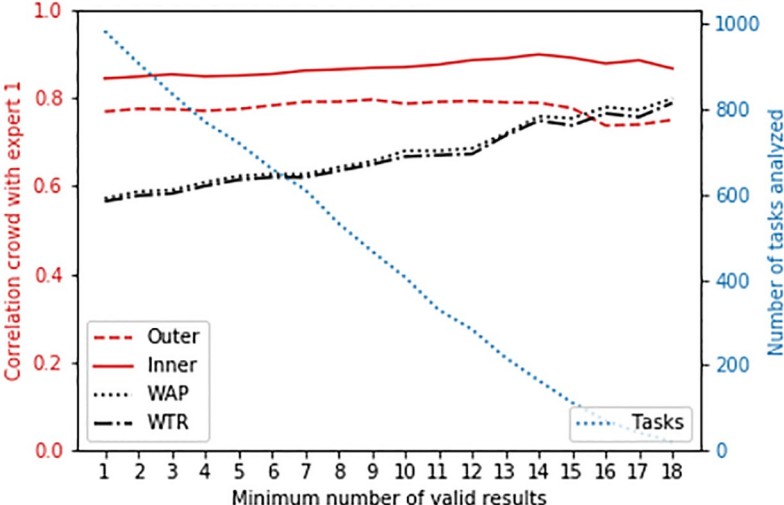

**Fig 9. Correlation of all four measures between the expert and crowd (median combining), for tasks with at least a specific number of valid results.**

combining strong correlations for the inner and outer airways (0.844, 0.769), but moderate to strong correlations for the ratios are observed (0.572, 0.565). It is important to note that a similar trend is noticeable in the expert-to-expert correlations: the correlations for the airway dimensions are much higher (0.964, 0.925) than correlations of WAP and WTR (0.701, 0.687).

### Predicting crowdsourcing quality

Next, we look at the correlations per subject, and whether this correlation can be predicted based on subject characteristics. The individual subject characteristics and correlations (between expert and median combining) are shown in Table 3. Overall we can see high variability across subjects. Correlations range between 0.64 and 1.00 for inner airway, 0.60 and 0.98 for outer airway, 0.07 and 0.75 for WAP, and 0.14 and 0.68 for WTR. Note that these correlations are based on smaller (and different) numbers of tasks (column "n").

Lastly we looked at the relationship between the quality of the crowd (here represented by the inner airway correlation) and five subject characteristics. The Spearman correlations and corresponding p-values are shown in Table 4. There is a weak negative correlation between the subject having CF and the crowd quality, however this correlation is not significant for the adjusted alpha level of 0.01. The other characteristics show almost no correlation with the crowd quality.

These results suggest that other factors, not investigated here, are more important. We suspect that these factors are related to the difficulty of individual tasks (for example depending on size, shape, and contrast of the airway and its proximity to vessels or other structures), and/or assignment of workers to different tasks.

### Discussion

This paper describes a follow-up study of [9]. In that study we concluded that workers try to annotate airways in the images but often do not create valid results. After filtering out the invalid results, the correlations between the crowd and the expert were 0.69 for the inner and 0.75 for the outer airway, and could be further improved by combining the results. As follow-

**Table 3. Characteristics of the subject: ID (not used in modeling), whether a subject has CF (1 = yes), FVC1, FEV (as percentage of predicted value), number of airways (n), and correlations between the crowd (median combining) and the expert.** Horizontal lines inserted for legibility.

| | Subject characteristics | | | | Correlation crowd-expert | | | |
|---|---|---|---|---|---|---|---|---|
| Index | CF | FVC1 | FEV | n | inner | outer | WAP | WTR |
| 0 | 1 | 109.20 | 119.20 | 77 | 0.93 | 0.97 | 0.52 | 0.42 |
| 1 | 0 | 111.50 | 95.40 | 89 | 0.93 | 0.98 | 0.33 | 0.36 |
| 2 | 1 | 112.20 | 99.00 | 60 | 0.64 | 0.92 | 0.07 | 0.14 |
| 3 | 0 | 82.70 | 78.60 | 168 | 0.78 | 0.94 | 0.35 | 0.36 |
| 4 | 0 | 110.00 | 105.10 | 73 | 0.90 | 0.90 | 0.61 | 0.59 |
| 5 | 1 | 118.60 | 94.50 | 68 | 0.76 | 0.71 | 0.75 | 0.75 |
| 6 | 0 | 95.60 | 85.40 | 173 | 0.90 | 0.87 | 0.45 | 0.43 |
| 7 | 1 | 94.20 | 73.40 | 82 | 0.91 | 0.95 | 0.45 | 0.43 |
| 8 | 0 | 99.40 | 73.20 | 143 | 0.75 | 0.97 | 0.39 | 0.37 |
| 9 | 0 | 120.30 | 123.80 | 122 | 0.93 | 0.97 | 0.60 | 0.61 |
| 10 | 1 | 70.60 | 75.30 | 75 | 0.82 | 0.86 | 0.22 | 0.21 |
| 11 | 0 | 104.20 | 100.10 | 134 | 0.77 | 0.90 | 0.50 | 0.51 |
| 12 | 1 | 73.40 | 56.10 | 47 | 0.91 | 0.91 | 0.51 | 0.44 |
| 13 | 1 | 70.60 | 78.10 | 18 | 0.65 | 0.60 | 0.58 | 0.59 |
| 14 | 0 | 98.90 | 95.90 | 145 | 0.93 | 0.92 | 0.56 | 0.56 |
| 15 | 0 | 73.40 | 66.70 | 278 | 0.87 | 0.93 | 0.43 | 0.39 |
| 16 | 1 | 128.60 | 80.60 | 66 | 0.81 | 0.98 | 0.42 | 0.45 |
| 17 | 1 | 91.90 | 79.10 | 40 | 0.86 | 0.98 | 0.37 | 0.35 |
| 18 | 1 | 96.90 | 91.10 | 27 | 1.00 | 0.89 | 0.48 | 0.53 |
| 19 | 1 | 109.00 | 105.80 | 104 | 0.94 | 0.98 | 0.67 | 0.68 |
| 20 | 0 | 110.10 | 104.10 | 32 | 0.88 | 0.98 | 0.54 | 0.55 |
| 21 | 1 | 64.30 | 69.60 | 64 | 0.87 | 0.94 | 0.24 | 0.26 |
| 22 | 0 | 109.60 | 82.00 | 151 | 0.86 | 0.93 | 0.42 | 0.44 |
| 23 | 0 | 77.00 | 71.20 | 144 | 0.91 | 0.94 | 0.63 | 0.61 |

**Table 4. Spearman correlation between the crowd quality (measured by the crowd-expert correlation of the inner airway) and five subject characteristics.**

| Characteristic | Correlation | p-value |
|---|---|---|
| Has CF | -0.265 | 0.211 |
| Generation | 0.003 | 0.987 |
| FVC | -0.038 | 0.859 |
| FEV1 | 0.090 | 0.677 |
| Number of airways: | -0.038 | 0.861 |

up steps, we revised our instructions to the crowd, increased the number of workers from 10 to 20 per slice, and collected annotations for all 24 subjects in the cohort.

The current study (increased from 1 to 24 subjects) shows that despite revising the instructions, the number of invalid annotations is still high (57.2%). This can happen when a worker does not see an airway, sees an airway but annotates it incorrectly, or due to spam (workers who submit random results just to get the reward). Our analysis shows that most workers create both valid and invalid results. An improvement to increase the number of valid results would be to perform checks (such as requiring one ellipse to be inside the other) inside the annotation interface.

After removing the invalid results, we examined the correlations between the crowd and the expert. Without combining, the correlations were strong for the inner and outer airway, and moderate for WAP and WTR. Combining results across tasks improved correlations, so that all four measures had strong correlations. However, all correlations were still lower than correlations between two experts (see Table 2).

We used simple combining methods to combine the results of the crowd. Here many alternatives are possible, such as weighting the workers by their estimated quality. Instead we tried to estimate an "upper limit" for the crowd with a method which selected the best available result for each task. This method did indeed lead to the highest correlations, but simple median combining was a close second. We conclude that median combining is suitable for this data.

Overall we conclude that the crowd is capable of producing good-quality, but not expert-quality, results. As such, in its current form the proposed method is not robust enough for gathering measurements "in the wild". In our experience this is primarily due to the difficulty of converting a clinical problem into a crowdsourcing problem, such as figuring out how to display parts of a 3D image in a 2D interface, explaining the task to the workers, and dealing with the constraints of the crowdsourcing platform.

Our study was inspired by the lack of annotated datasets for machine learning, however, we focused on evaluating the quality of the annotations alone. There are indications that lower quality labels may still be useful for training machine learning algorithms. For example, combinations of expert and crowdsourced labels have shown to be more effective in cases where the crowdsourced labels alone were not sufficient [14]. More generally, weakly-supervised learning with approximate annotations such as bounding boxes shows that labels do not need to be precise to add value during training [15].

There are a number of important lessons from this study, which could be valuable for other researchers doing similar studies. Firstly, our interface was custom built by a crowdsourcing start-up, in the context of a pilot for academic groups. This allowed us to use an ellipse tool that is similar to the tool used by the experts. A disadvantage of this approach is that we could not easily access the interface after the pilot ended, and thus would not be able to collect additional data.

Secondly, although we did a test run of the task (collecting results described in [9]), we did not gather feedback from the workers about their experiences. This is possible through various online groups such as https://www.turkernation.com, and could have reduced possible misunderstandings of the instructions. Furthermore, we used crowd qualifications (such as acceptance rate) and rewards that are outdated by today's standards, so we would recommend other researchers to consult the latest crowdsourcing literature before setting up such a study.

For reproducibility of the results and any follow-up analyses, we made the airway images, crowd results and our code available via http://github.com/adriapr/crowdairway.git.

## Conclusions

We conducted a follow-up study of crowdsourcing airway measurements in slices of chest CT scans. In a previous, preliminary study we examined airways of one subject and found moderate to strong correlations with the expert, motivating this follow-up study. Overall we observed similar or better results, with strong correlations for the inner and outer airway dimension measurements, and moderate to strong correlations for ratios of these structures. Combining results across different workers improved the correlations, but correlations were still lower than between two experts. We conclude that with appropriate

processing, the crowd has potential in measuring airways, but that our method is not yet robust enough for use in practice.

## Author Contributions

**Conceptualization:** Veronika Cheplygina, Marleen de Bruijne.

**Data curation:** Wieying Kuo, Harm A. W. M. Tiddens.

**Formal analysis:** Veronika Cheplygina, Adria Perez-Rovira.

**Funding acquisition:** Veronika Cheplygina, Harm A. W. M. Tiddens, Marleen de Bruijne.

**Investigation:** Veronika Cheplygina.

**Methodology:** Veronika Cheplygina, Adria Perez-Rovira, Marleen de Bruijne.

**Project administration:** Veronika Cheplygina.

**Resources:** Wieying Kuo, Harm A. W. M. Tiddens, Marleen de Bruijne.

**Software:** Veronika Cheplygina, Adria Perez-Rovira.

**Supervision:** Marleen de Bruijne.

**Validation:** Veronika Cheplygina, Adria Perez-Rovira.

**Visualization:** Veronika Cheplygina, Adria Perez-Rovira.

**Writing – original draft:** Veronika Cheplygina, Adria Perez-Rovira.

**Writing – review & editing:** Veronika Cheplygina, Adria Perez-Rovira, Marleen de Bruijne.

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
