## [Decision Letter · Decision Letter 0]

5 Feb 2021

PONE-D-20-37113

Crowdsourcing Airway Annotations in Chest Computed Tomography Images

PLOS ONE

Dear Dr. Cheplygina,

Thank you for submitting your manuscript to PLOS ONE. After careful consideration, we feel that it has merit but does not fully meet PLOS ONE’s publication criteria as it currently stands. Therefore, we invite you to submit a revised version of the manuscript that addresses the points raised during the review process.

We look forward to receiving your revised manuscript.

Kind regards,

Ulas Bagci, Ph.D.

Academic Editor

PLOS ONE

Additional Editor Comments (if provided):

The paper has a considerably interest in the field, easy to follow, with some minor concerns raised by the reviewers.

Journal Requirements:

2) Please provide additional details regarding participant consent. In the Methods section, please ensure that you have specified how consent was obtained and how the study met relevant personal data and privacy laws. If data were collected anonymously, please include this information.

3) Please include a copy of Tables 1 & 2 which you refer to in your text.

Reviewers' comments:

Reviewer's Responses to Questions

**Comments to the Author**

1. Is the manuscript technically sound, and do the data support the conclusions?

Reviewer #1: Yes

Reviewer #2: Yes

2. Has the statistical analysis been performed appropriately and rigorously? 

Reviewer #1: Yes

Reviewer #2: Yes

3. Have the authors made all data underlying the findings in their manuscript fully available?

Reviewer #1: Yes

Reviewer #2: Yes

4. Is the manuscript presented in an intelligible fashion and written in standard English?

Reviewer #1: Yes

Reviewer #2: Yes

5. Review Comments to the Author

Reviewer #1: This paper presents an experiment of crowdsourcing for medical image annotation, specifically annotating airway and airway wall from 2D resampled CT scans. I would say this is a valid study and whether crowdsourcing can be used to for medical applications is an interesting topic. Overall the manuscript is clear and easy to follow.

Some details that need to be considered in revision:

- “is our earlier study [2] where we described our experiences with crowdsourcing airway measurements.” [2] does not seem to be the correct citation, guess it should be [9]?

- What is the voxel size of the CT images? What is the physical length of 50x50 voxels?

- The 1026 image patches, how are they distributed along the airway tree? Which generation of branch do they belong to? It will help to understand the approximate size of the airways, by comparing with the dimensions from the last question.

- Inner and outer diameter, since they are ellipses, which “diameter” is used? Major? Minor? Average of major and minor? or average of all?

- Just to confirm, considering the “not seeing an airway” cases, for the input image patches, airway is always visible right? Is there a case that everyone marks as “not seeing”?

- Table 2, Expert 1 and 2 seems to have fairly good agreement, still, it may be helpful to also give the coor number for three combination against Expert 2 (and maybe further against the average number of 1&2)

- Maybe out-of-scope for this paper, but it would be interesting to see how good a SOTA automated algorithm can achieve on the same samples.

Reviewer #2: The authors facilitated Amazon Mechanical Turk to crowdsource airway annotations from chest CT slices. Annotations from multiple workers were analyzed against the annotations provided by two experts. Almost half of the annotations from workers were identified as invalid due to insufficient instructions and/or issues in experimental design. These issues are discussed in the manuscript and the manuscript provides enough information to design a robust crowdsourcing platform for chest CT airway annotations.

The authors used crowdsourcing to annotate datasets to machine learning algorithms, as indicated in the abstract. However, experiments analyzing the effect of using annotated dataset generated by workers in ML algorithms were not performed or discussed. It is not clear if nonexpert annotated “good-quality” dataset can be of any use in ML model development or not. Please elucidate.

Page 7 2nd paragraph of the Discussion Section. Please add the “%” sign after 57.2

The manuscript is easy to follow, with issues on the Figure numbers which could be mitigated by using the Figure number as a filename.

6. PLOS authors have the option to publish the peer review history of their article (what does this mean?). If published, this will include your full peer review and any attached files.

Reviewer #1: No

Reviewer #2: No

---

## [Author Response · Author response to Decision Letter 0]

12 Feb 2021

Thank you for taking the time to review our manuscript. Please view response_letter.pdf for our point by point response.

---

## [Decision Letter · Decision Letter 1]

22 Mar 2021

Crowdsourcing Airway Annotations in Chest Computed Tomography Images

PONE-D-20-37113R1

Dear Dr. Cheplygina,

We’re pleased to inform you that your manuscript has been judged scientifically suitable for publication and will be formally accepted for publication once it meets all outstanding technical requirements.

Kind regards,

Ulas Bagci, Ph.D.

Academic Editor

PLOS ONE

Additional Editor Comments (optional):

Authors successfully answered the comments.

Reviewers' comments:

Reviewer's Responses to Questions

**Comments to the Author**

1. If the authors have adequately addressed your comments raised in a previous round of review and you feel that this manuscript is now acceptable for publication, you may indicate that here to bypass the “Comments to the Author” section, enter your conflict of interest statement in the “Confidential to Editor” section, and submit your "Accept" recommendation.

Reviewer #1: All comments have been addressed

2. Is the manuscript technically sound, and do the data support the conclusions?

Reviewer #1: Yes

3. Has the statistical analysis been performed appropriately and rigorously? 

Reviewer #1: Yes

4. Have the authors made all data underlying the findings in their manuscript fully available?

Reviewer #1: Yes

5. Is the manuscript presented in an intelligible fashion and written in standard English?

Reviewer #1: Yes

6. Review Comments to the Author

Reviewer #1: (No Response)

7. PLOS authors have the option to publish the peer review history of their article (what does this mean?). If published, this will include your full peer review and any attached files.

Reviewer #1: No

---

## [Editor Report · Acceptance letter]

31 Mar 2021

PONE-D-20-37113R1 

Crowdsourcing Airway Annotations in Chest Computed Tomography Images 

Dear Dr. Cheplygina:

I'm pleased to inform you that your manuscript has been deemed suitable for publication in PLOS ONE. Congratulations! Your manuscript is now with our production department. 

Kind regards, 

on behalf of

Dr. Ulas Bagci 

Academic Editor

PLOS ONE